# Efficient phonon cascades in WSe$_2$ monolayers

Ioannis Paradisanos [1,2✉], Gang Wang[2,3], Evgeny M. Alexeev [2], Alisson R. Cadore[2], Xavier Marie [1], Andrea C. Ferrari [2✉], Mikhail M. Glazov [4✉] & Bernhard Urbaszek [1✉]

Energy relaxation of photo-excited charge carriers is of significant fundamental interest and crucial for the performance of monolayer transition metal dichalcogenides in optoelectronics. The primary stages of carrier relaxation affect a plethora of subsequent physical mechanisms. Here we measure light scattering and emission in tungsten diselenide monolayers close to the laser excitation energy (down to ~0.6 meV). We reveal a series of periodic maxima in the hot photoluminescence intensity, stemming from energy states higher than the A-exciton state. We find a period ~15 meV for 7 peaks below (Stokes) and 5 peaks above (anti-Stokes) the laser excitation energy, with a strong temperature dependence. These are assigned to phonon cascades, whereby carriers undergo phonon-induced transitions between real states above the free-carrier gap with a probability of radiative recombination at each step. We infer that intermediate states in the conduction band at the $\Lambda$-valley of the Brillouin zone parti- cipate in the cascade process of tungsten diselenide monolayers. This provides a fundamental understanding of the first stages of carrier–phonon interaction, useful for optoelectronic applications of layered semiconductors.

[1] Université de Toulouse, INSA-CNRS-UPS, LPCNO, 135 Avenue Rangueil, Toulouse 31077, France. [2] Cambridge Graphene Centre, University of Cambridge, Cambridge CB3 0FA, UK. [3] Key Lab of Advanced Optoelectronic Quantum Architecture and Measurement (MOE), School of Physics, Beijing Institute of Technology, Beijing 100081, China. [4] Ioffe Institute, St.-Petersburg 194021, Russia. ✉email: paradeis@insa-toulouse.fr; acf26@eng.cam.ac.uk; glazov@coherent.ioffe.ru; urbaszek@insa-toulouse.fr

The optical properties of group VI transition metal dichalcogenide monolayer (1L-TMD) semiconductors are dominated by excitons (bound electron-hole, e-h, pairs) with binding energies of hundreds of meV[1], with spin and valley properties (such as valley-selective circular dichroism[2]) highly beneficial for optoelectronics[2], valleytronics[3] and spintronics[3–12]. Following optical excitation of a semiconductor above the band gap, the subsequent energy relaxation pathways play an important role in optics[13–15] and charge carrier transport[16,17]. These processes are related to hot (i.e. not in thermal equilibrium) charge carriers and excitons[1], and determine electron mobility[18], optical absorption in indirect band gap semiconductors[19], and intervalley scattering of hot electrons[19]. Photoluminescence (PL) and Raman scattering can be used to probe the interactions of carriers with phonons[20]. Different types of phonons with different energies can participate in the relaxation process of excited carriers. However, in some materials one type of phonon plays a dominant role and leads to high-order processes, e.g. up to nine longitudinal optical (LO) phonon replicas were reported in the hot PL of CdS and CdSe[20–22]. Multiphonon processes are important in defining the optoelectronic performance of ZnO[23–26], GaN[27] and bulk MoS$_2$[28]. The optical oscillator strength in 1L-TMDs, i.e. the probability of optical transitions between valence and conduction states, is higher than in III-V quantum wells[19], resulting in short (~1ps[29]) exciton lifetimes. This favors hot PL emission, as excitons relax between several real states[30,31]. Examination of phonon-induced cascade-like relaxation processes in 1L-TMDs has been proposed for future pump-probe experiments[32]. However, observation of direct optical signatures in the early stages of carrier relaxation still remains a significant challenge, because of the ultrafast timescale (~100fs[33]) of these processes. Understanding the relaxation pathways in tungsten diselenide monolayers (1L-WSe$_2$) is

important for optoelectronic applications, such as photo-detectors[34] and lasers[35], because it determines the recovery rate (i.e. the population of carriers relaxing to the ground state over time) and, as a result, the devices' speed and efficiency.

Here, we use ultra-low (~5 cm$^{-1}$ ~0.6 meV) cut-off frequency (ULF) Raman spectroscopy to investigate the light scattered and emitted by 1L-WSe$_2$ on SiO$_2$, hBN and Au, as well as suspended 1L-WSe$_2$. We observe phonon-assisted emission of hot PL, periodic in energy both in the Stokes (S) and anti-Stokes (AS) spectral range, and we extract a phonon energy ~15 meV. The S signal shows 7 maxima in the range of temperatures (T) from 78 to 295 K. We also detect up to 5 maxima in the AS signal ~75 meV above the laser excitation energy, increasing in intensity as T is raised. We assign these to phonon cascades[36]. We include finite T effects to compare S and AS signals and to understand carrier relaxation at room temperature (RT). By analyzing the T and excitation energy dependence, we conclude that a continuum of states (in the free-carrier gap) is involved in e-h relaxation in 1L-WSe$_2$. Intermediate states in the conduction band around the Λ-valley of the Brillouin zone (BZ) participate in the cascade process. Hot PL so close in energy to the excitation laser gives access to the initial stages of carrier relaxation. These processes are ultrafast (e.g. ~100fs in GaAs[33]) and it is therefore challenging to access them in time-resolved experiments. Our approach can be extended to all layered materials (LMs) and their heterostructures (LMHs), as well as to other materials systems, such as perovskites[37,38].

## Results

1L-WSe$_2$ flakes are exfoliated from bulk 2H-WSe$_2$ crystals (2D Semiconductors) by micromechanical cleavage on Nitto Denko tape[39], then exfoliated again on a polydimethylsiloxane (PDMS)

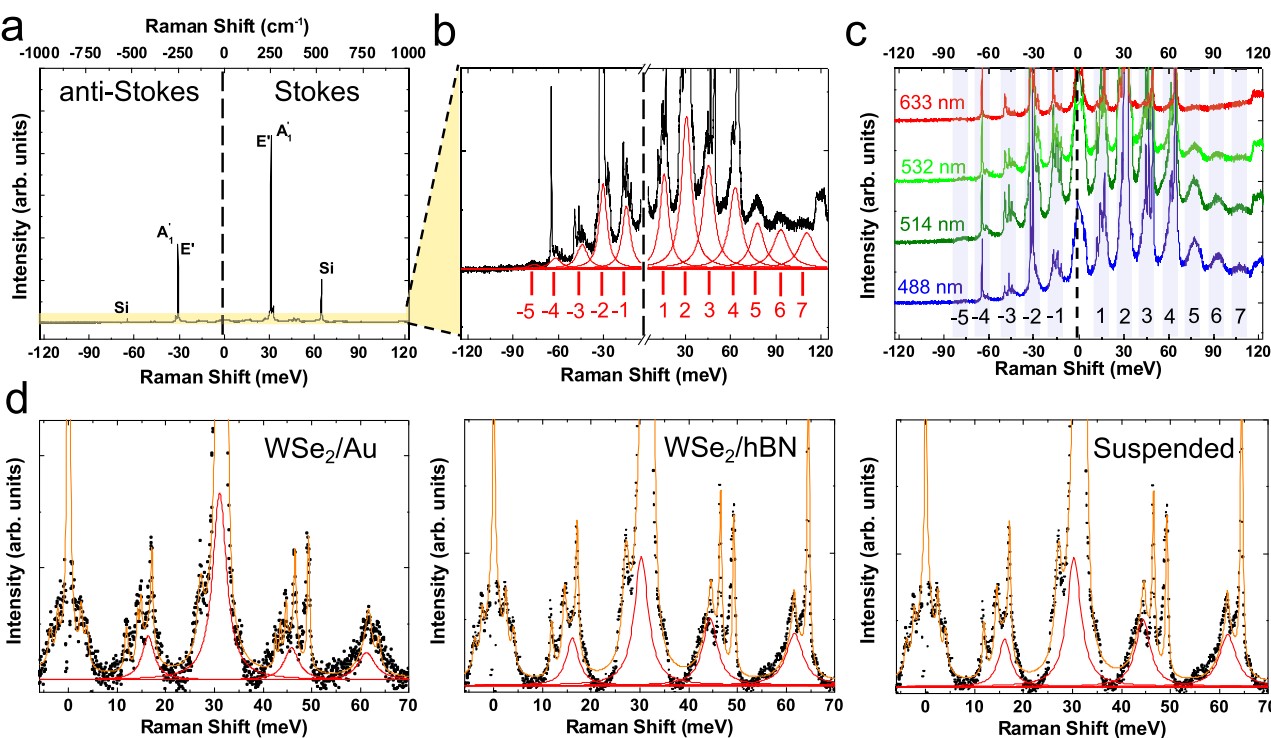

**Fig. 1 Raman and hot PL spectra of 1L-WSe$_2$. a** Emission and scattering spectrum of 1L-WSe$_2$ at 295K as a function of energy shift with respect to the excitation laser (532 nm~2.33eV). The degenerate in-plane ($E'$) and out-of-plane ($A'_1$) Raman mode ~250 cm$^{-1}$ [43], as well as the Si Raman peak ~521 cm$^{-1}$ [47], are prominent in both S and AS. **b** Magnified portion of the spectrum in yellow in **a**. This reveals 7 periodic S peaks and 5 AS. Their intensity decreases as a function of the energy shift for both S and AS. **c** Raman spectra of 1L-WSe$_2$ on SiO$_2$/Si at 488, 514, 532, 633 nm and 295 K, shifted vertically for clarity. **d** Raman spectra of 1L-WSe$_2$ on different substrates (Au, hBN and suspended) at 295K and 514 nm excitation. Black points: experimental data. Red lines: fitted cascades. Orange line: sum of fitted Lorentzians.

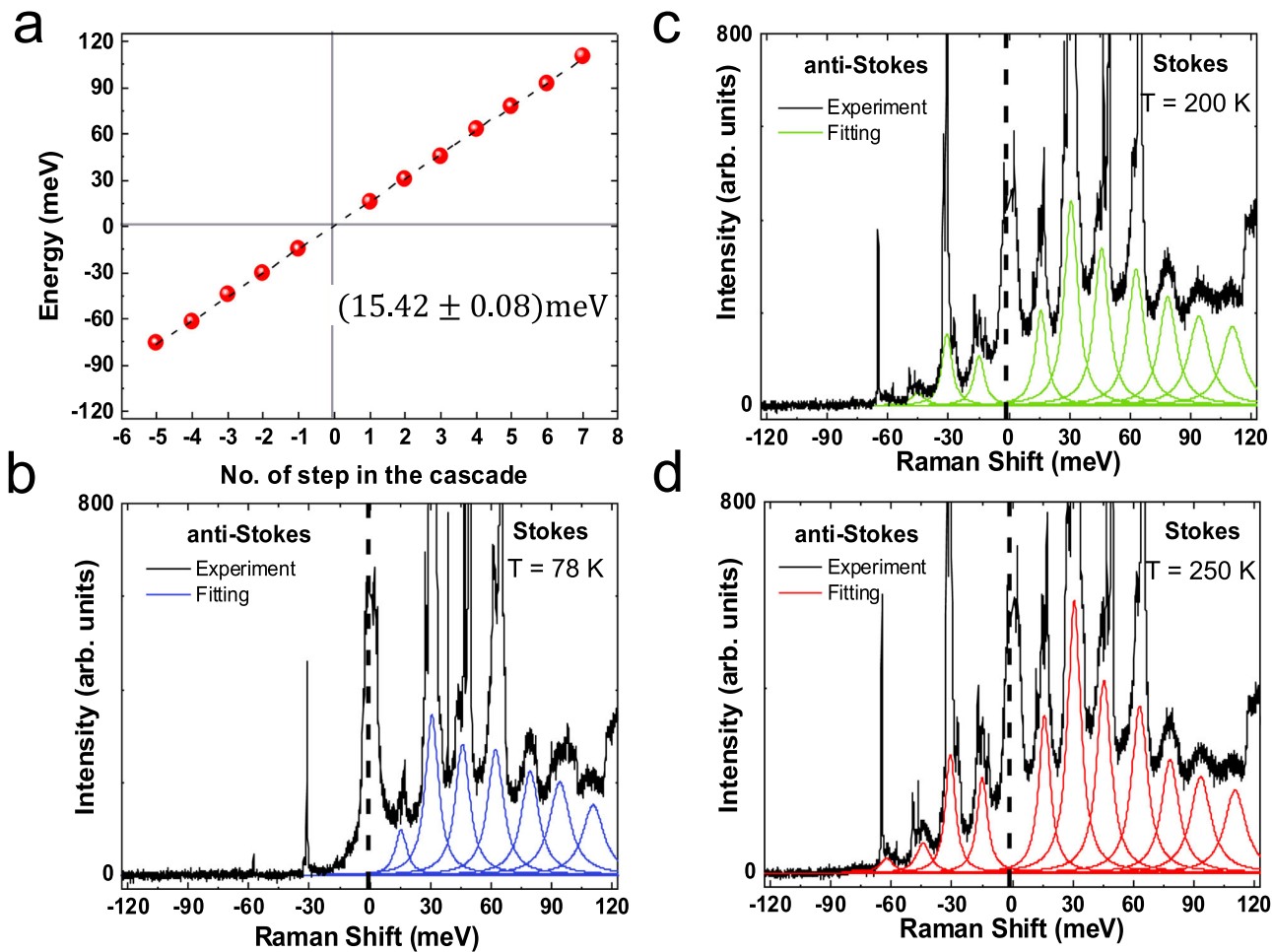

**Fig. 2 Energy separation and T dependence. a** Emission energies as a function of number of steps in the cascade, extracted from the RT spectrum in Fig. 1b. The dashed black line is a linear fit, giving a step energy ~15.42 ± 0.08 meV. **b–d** 532 nm Hot PL spectra of 1L-WSe$_2$ at **b** 78 K, **c** 200 K, **d** 250 K.

stamp placed on a glass slide for inspection under optical microscope. Optical contrast is used to identify 1L prior to transfer[40]. Before transfer, 85 nm (for optimum contrast[40]) SiO$_2$/ Si substrates are wet cleaned[41] (60s long ultrasonication in acetone and isopropanol) and subsequently exposed to oxygen-assisted plasma at 10W for 60s. The 1L-WSe$_2$ flakes are then stamped on the substrate with a micro-manipulator at 40 °C, before increasing T up to 60 °C to release 1L-WSe$_2$[42]. The same procedure is followed for transfer of 1L-WSe$_2$ on hBN, Au and Si, with 2 μm Au trenches made by lithography, to suspend the samples.

The Raman and hot PL spectra are recorded in a back-reflection geometry with a ×50 objective (NA = 0.45) and a spot size ~1 μm. A liquid nitrogen cryostat (Linkam Scientific) placed on a XY translational stage is used to control T between 78 K and 295 K and excitation area. Imaging of the sample and monitoring of the excitation spot position are achieved using a set of beam splitters, aligned to a charge-coupled device (CCD) camera. The PL and Raman signals collected in the backward direction are filtered by 3 notch volume Bragg filters with a total optical density (OD) = 9. The cut-off frequency is ~5 cm$^{-1}$ (~0.6 meV). The filtered signals are then focused on the spectrometer slit and dispersed by a 1800l/mm grating before being collected by the detector.

A typical RT Raman spectrum for 1L-WSe$_2$ on SiO$_2$/Si measured at 532 nm is shown in Fig. 1a. The degenerate in-plane, $E'$, and out-of-plane, $A_1'$, modes of 1L-WSe$_2$[43] dominate the spectrum

at ~−250 cm$^{-1}$ (−31 meV) and ~+250 cm$^{-1}$ (+31 meV) in the AS and S range, while weaker Raman peaks are also observed between 90 cm$^{-1}$ (11 meV) and 500 cm$^{-1}$ (62 meV) (see Methods and Supplementary Note 1) as discussed in refs. [44–46]. Rescaling the intensity within the region marked in yellow in Fig. 1a reveals an underlying periodic pattern, Fig. 1b. Hereafter, for the energy scale we will use meV instead of cm$^{-1}$. We fit all the peaks between −120 meV and +120 meV using Lorentzians, as shown in red in Fig. 1b. The fitting process is described in Methods. There are 7 S peaks and 5 AS at 295 K. The peak ~120 meV (~970 cm$^{-1}$) originates from a combination of the Si substrate $\Gamma_1, \Gamma_{12}, \Gamma_{25'}$ phonons[47]. Although the energy separation between two consecutive peaks is constant, the intensity decreases as a function of energy with respect to the excitation energy (here fixed at 0). To exclude other contributions, such as thin-film interference effects[48], we measure 1L-WSe$_2$ transferred on Au, placed on top of few-layer (FL) (~10 nm) hBN and also suspended, Fig. 1d (see Methods and Supplementary Note 1 for optical microscope images and PL characterisation). The intensity of the hot PL is comparable among the same steps of the cascade, and the position of the peaks is the same. Therefore, the cascade is linked to intrinsic relaxation mechanisms of 1L-WSe$_2$, not to substrate-induced interference. Henceforth we will focus on 1L-WSe$_2$ on SiO$_2$/Si.

To exclude the possibility that our laser is in resonance with a specific transition, we perform variable excitation wavelength experiments at 295K. Figure 1c plots the spectra measured at

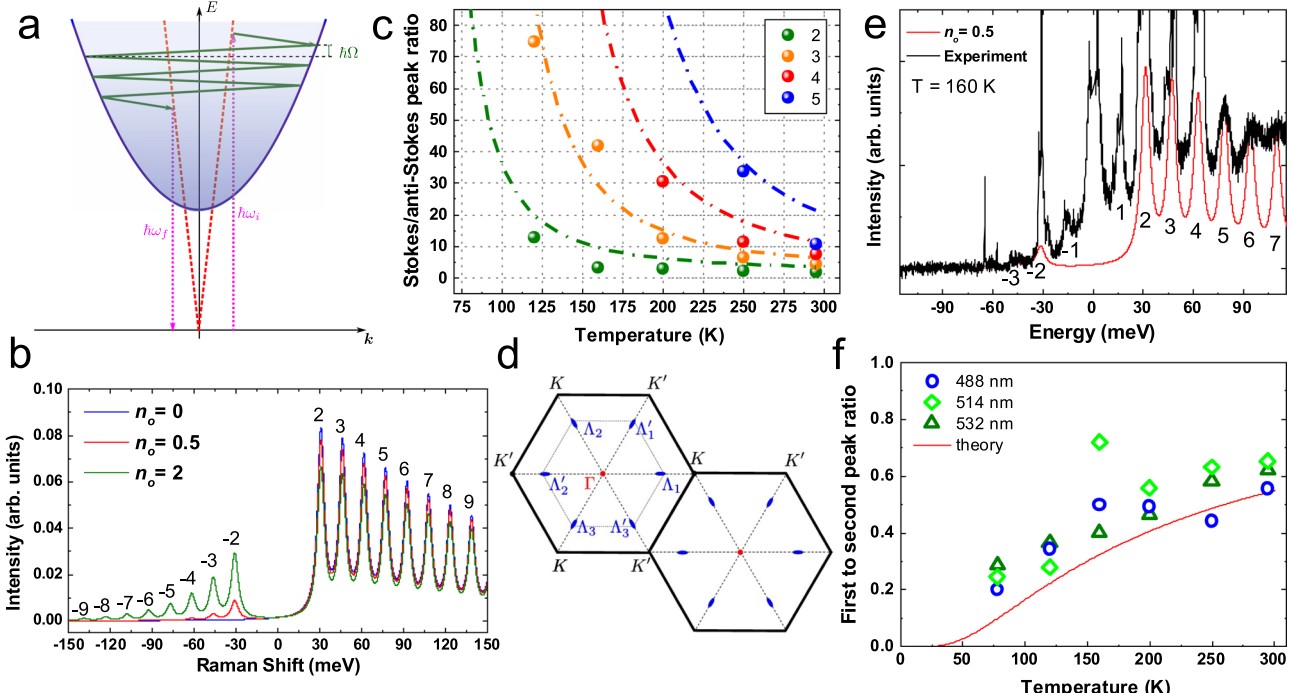

**Fig. 3 Comparison between experiments and theory. a** Scheme of phonon-assisted hot PL. The incident, $\hbar\omega_i$, and outgoing, $\hbar\omega_f$, photons are shown by dotted magenta vertical arrows. The phonons participating in the cascade are indicated by the green arrows. The e–h pair dispersion curve is the blue parabola. The light cone is shown by red dashed lines. **b** Calculated S/AS spectra at different T. **c** $I_S/I_{AS}$ for different numbers of cascade steps as a function of T. Filled circles are experimental data at 532 nm. The fit with Eq. (9) is indicated by dot-dashed lines. **d** Extended BZ of 1L-WSe2. Corresponding valleys are marked as $\Gamma$, $K$, $K'$ and $\Lambda_i$, $\Lambda'_i$ ($i = 1, ..., 3$). **e** Experimental data at 160K (black line) compared to the calculated spectrum from Eq. (2) for $n_o = 0.5$. **f** Ratio of measured intensities of $j = 1$ to $j = 2$ peaks and corresponding fit with Eq. (10).

488 nm (~2.54 eV), 514 nm (~2.41 eV), 532 nm (~2.33 eV) and 633 nm (~1.96 eV). We observe the same high-order features with identical energy separations in both S and AS. All these excitation energies lie above the free-carrier gap of 1L-WSe2 ~ 1.89 eV[49–51]. By comparing results for 1L-WSe2 on different substrates and for different excitation energies, we deduce that phonon-assisted hot PL is the dominant mechanism, whereas contributions from other excitations, such as plasmons[52], are negligible, otherwise intensity and/or energy variations would be expected between Au and SiO2/Si, hBN, suspended cases.

Figure 2a plots the energy offset with respect to the excitation laser (here 532 nm) of each emission feature as a function of the number of steps in the cascade at 295 K. Applying a linear fit, we extract ~15.42 ± 0.08 meV, regardless of substrate and excitation energy. This periodic modulation of the detected light intensity suggests that the scattering of photoexcited carriers is dominated by one prominent phonon mode. Since we excite above the free-carrier gap of 1L-WSe2[49], the intermediate states of the transitions are real. The e-h pair representation is in Fig. 3a.

The lattice T could affect the peaks intensity, as phonon occupation increases with T[53,54]. We thus perform T dependent measurements from 78 to 295 K, while keeping the excitation power constant ~26 μW. No emission of AS features is observed at 78 K, Fig. 2b, with the exception of two sharp lines ~−30 and ~−60 meV, originating from 1L-WSe2 and Si Raman modes, respectively. The hot PL peaks are seen at 200 K, Fig. 2c, and a further increase in intensity is observed at 250 K, Fig. 2d. Additional measurements at 120, 160, and 295 K are performed and used in the fits in Fig. 3c. Thermal effects are expected to modify the phonon energies[55]. However, in the 78–298 K range we do not observe any measurable shifts in the position of the hot PL peaks, because the shifts induced by acoustic phonons are smaller compared to our experimental error, as discussed in Methods.

## Discussion

At low T (78 K), phonon absorption processes are suppressed because of the insufficient lattice thermal energy[53]. Optical excitation results in free e-h pair formation[56,57] or virtual formation of an exciton with small in-plane wavevector ($\mathbf{k} \lesssim \omega_i/c$ with $\omega_i$ the excitation laser frequency)[1]. With the subsequent phonon emission, the e-h pair reaches a real final state (blue parabola in Fig. 3a), for which radiative recombination is forbidden by momentum conservation[19]. This triggers a cascade relaxation process[36], whereby at each step a phonon is emitted (or absorbed for a T whereby the thermal energy is equal or higher than the phonon one energy)[19]. If the interaction with one phonon mode with energy $\hbar\Omega$ dominates overall other inelastic scattering processes, the exciton loses energy by integer multiples of $\hbar\Omega$[19,36]. After emission of several (≥2) phonons, the exciton recombines and emits a photon with frequency $\omega_f$ in a two-step process via an intermediate state with a small ($\mathbf{k} \lesssim \omega_i/c$) wavevector, for which radiative recombination is momentum allowed. Thus, we have secondary emission or scattering of light with S shift $\omega_i - \omega_f = j\Omega$, where $j = 2, 3, ....$, while $j = \pm 1$ are impossible as we scatter out of the light cone (i.e. the region of small wavevectors) with the first event. At finite T, in addition to phonon emission, absorption also comes into play, and AS emission is observed at $\omega_f - \omega_i = j\Omega$.

Multiphonon processes that do not involve real states require higher order exciton–phonon interactions[58], and are therefore less probable. In contrast, the process in Fig. 1c is resonant, since excitation in the free-carrier gap means that all intermediate states are real. This allows us to describe the phonon emission cascade via the kinetic equation for the exciton distribution function $f(\varepsilon)$, where $\varepsilon$ is the exciton energy, as derived in Supplementary Notes 2, 3. Since the energy of the exciton changes in

each scattering event by $\pm\hbar\Omega$, the distribution function can be written as:

$$f(\varepsilon) = \sum_{j=-\infty}^{\infty} f_j \delta(\varepsilon_0 - j\hbar\Omega) \qquad (1)$$

where $\varepsilon_0$ is the excitation energy, $\delta(\varepsilon)$ is the Dirac $\delta$-distribution (phonon dispersion and damping result in the broadening of the $\delta$-distribution, as detailed in Supplementary Notes 2, 3), $f_j$ describes the peaks intensity. At steady state (partial derivative with respect to time equals zero) these obey a set of coupled equations describing the interplay of in- and out-scattering processes:

$$\gamma f_j = \gamma_o \left[ f_{j-1}(n_o + 1) + f_{j+1}n_o \right] + g\delta_{j,0}, \qquad (2)$$
$$j = \dots, -2, -1, 0, 1, 2, \dots .$$

where $n_o = \left[ \exp(\hbar\Omega/k_B T) - 1 \right]^{-1}$ is the phonon mode occupancy at $T$, $\gamma_o$ is the rate of the spontaneous phonon emission, $\gamma = \gamma_o(2n_o + 1) + \gamma'$, is the total damping rate of the exciton, which includes recombination and inelastic scattering processes $\gamma'$. The last term in Eq. (2), $g\delta_{j,0}$, describes the exciton generation at energy $\varepsilon_0$, and is proportional to the exciton generation rate. Eq. (2) has the boundary conditions:

$$\lim_{j \to -\infty} f_j = 0, \quad f_{K+1} = 0, \qquad (3)$$

where $K$ is the maximum number of steps in the cascade:

$$K = \left\lfloor \frac{\hbar\omega_i - E_1}{\hbar\Omega} \right\rfloor, \qquad (4)$$

with $E_1$ the energy of the exciton band bottom. Eq. (2) is derived assuming $\gamma_o$ and $\gamma'$ independent of $\varepsilon$. This assumption is needed to get an analytical solution of Eq. (2), but can be relaxed (see Supplementary Notes 2, 3).

The general solution of Eq. (2) is:

$$f_j = \begin{cases} A x_+^j, & j > 0, \\ B x_+^j + C x_-^j, & j \le 0, \end{cases} \qquad (5)$$

where

$$x_{\pm} = \frac{\gamma \pm \sqrt{\gamma^2 - 4n_o(n_o + 1)\gamma_o^2}}{2\gamma_o n_o}, \qquad (6)$$

and $x_+ > 1$ and $x_- < 1$, $A$, $B$ and $C$ are the coefficients. For cascades with $K \gg 1$ we can set $B = 0$ and:

$$A = C = \frac{g}{\sqrt{\gamma'^2 + 2\gamma_o\gamma'(1 + 2n_o)}}. \qquad (7)$$

In this model, the spectrum of the scattered light consists of peaks with $I \propto f_j$, and scattering cross-section:

$$\sigma(\omega_i, \omega_f) = \sigma_0(\omega_i, \omega_f)$$
$$\times \sum_{j=2}^{\infty}{}' \frac{1}{\pi} \frac{2\Gamma}{4\Gamma^2 + (j\Omega - \omega_i + \omega_f)^2} f_j. \qquad (8)$$

Here $\sigma_0(\omega_i, \omega_f)$ is a smooth function of frequency, $\Gamma$ is the phonon damping. This description is valid for peaks with $|j| > 1$, the prime at the summation denotes that the terms with $j = 0, \pm 1$ are excluded. Accordingly, the peaks with Raman shift $\pm\hbar\Omega$ are suppressed. At $n_o \to 0$ (limit of low T), $x_+ \gg 1$ and $I_j$ with negative $j$ (AS components) are negligible. At the same time, $x_- \to (\gamma_o/\gamma)$ and the S peak intensities, $I_S$, scale as $(\gamma_o/\gamma)^j$. This scaling is natural for cascade processes[19,59,60], since the probability of phonon emission relative to all other inelastic processes is given by $\gamma_o/\gamma$, thus $I_S$ decays in geometric progression. At finite T, the AS peaks appear with $I_{AS}$ proportional to the thermal occupation of the phonon modes. Thus, the S/AS intensity ratio,

$I_S/I_{AS}$, with $j$ steps in the cascade, can be written as:

$$\frac{I_S(j)}{I_{AS}(j)} = \frac{f_j}{f_{-j}} = \left( 1 + \frac{1}{n_o} \right)^j, \qquad (9)$$

and corresponds to the ratio of phonon emission and absorption rate to the power of $j$.

The calculated I distribution and spectra at various T (corresponding to different $n_o$) are presented in Fig. 3b. Figure 3c plots $I_S/I_{AS}$ as a function of T from Eq. (9). The experimental points collected from the fitted I of each step in the cascade at 532 nm excitation are displayed with circles. The absence of data at 78 K indicates no detection of $I_{AS}$ at this T. Applying Eq. (9) to the steps 2–5 in the cascade, with a phonon energy ~15.4 meV extracted from Fig. 2a, gives the dashed lines in Fig. 3c, in good agreement with experiments.

Our model captures the main experimental observations well. The periodic pattern of hot PL intensity is reproduced by the calculations, Fig. 3b, and $I_S/I_{AS}$ closely follows Eqs. (9), Fig. 3c. There is good agreement between our data and the calculated spectra from Eq. (2). An example for $n_o = 0.5$ at 160 K is in Fig. 3e. In our model, the peaks with $j = \pm 1$ are absent because $N \ge 2$ phonons are needed for the first step of the cascade process, as for Fig. 3a. However, Fig. 1 shows that $j = \pm 1$ peaks are smaller than $j = \pm 2$ ones, but still detectable. We consider $I_S(1)/I_S(2)$ as plotted in Fig. 3f. The possible mechanisms of $j = 1$ peak formation are as follows. (i) Elastic disorder or acoustic phonon-induced scattering, which provides a transfer between states within the light cone and states at the dispersion. (ii) Combination of phonon emission and absorption, where the $j = 1$ peak appears as a result of two phonon emission, followed by one phonon absorption. In (i) $I_S(1)/I_S(2)$ does not depend on T. In (ii):

$$\frac{I_S(1)}{I_S(2)} = \exp\left( -\frac{\hbar\Omega}{k_B T} \right), \qquad (10)$$

strongly depends on T. This is indeed the case in our experiment, see Fig. 3f. This additional channel is also based on the interaction with the same phonon energy ~15 meV. Elastic processes could be the origin of a small offset between the experiment and the fitted curve.

To get a better understanding of the relaxation pathways, we consider different scattering mechanisms. Scattering within the same valley is not plausible, due to the mismatch of BZ centre phonon energies[61]. The energy ~15 meV could correspond to either $\Gamma - K$ or $\Gamma - \Lambda$ phonons. The phonon dispersion in 1L-WSe$_2$ shows acoustic phonons with energies ~15 meV[46,61]. These have a flat dispersion, necessary to observe the high number of oscillations we report, and are compatible with the model in Fig. 3a.

Another option involves $K$-$K'$ scattering of e (h) or, equivalently, $\Gamma$-$K$ scattering of excitons. This would result in intensity oscillations as a function of the step in the cascade, due to the suppression of the process $\Gamma \to K \to K' \to \Gamma$ compared to, $\Gamma \to K \to \Gamma$ (see Supplementary Notes 4, 5). However, we do not observe intensity oscillations for different cascade steps in our spectra. As a result, we exclude this scenario. Therefore, the excitonic states in the $\Lambda$ valleys play a role as intermediate states, Fig. 3d. The conduction band minima in these valleys are relatively close (~35 meV) to $K$, and play a crucial role in exciton formation and relaxation[62–65]. In this case, h remain in $K$ (or $K'$), but e scatter to any of the 6 available $\Lambda$ valleys, and then scatter between these $\Lambda$ valleys, before going back to $K$ ($K'$). This can be

described taking into account all pathways, as:

$$\text{photon} \rightarrow \Gamma \xrightarrow{\hbar\Omega} \underbrace{\Lambda_i \xrightarrow{\hbar\Omega} \dots \xrightarrow{\hbar\Omega} \Lambda_j'}_{j} \xrightarrow{\hbar\Omega} \text{photon},$$

with arbitrary number of steps $j$ (both odd and even). The matrix elements of the processes are similar. We do not observe any noticeable periodic emission for 1L-MoS$_2$ and 1L-WS$_2$. This supports our interpretation, as the phonon scattering mechanism is linked to the particular bandstructure of 1L-WSe$_2$[62].

Similar oscillations can appear for free e and h[36], see Supplementary Note 4. The basic description of the effect is similar to what we observe here, and our model can be extended to take into account e/h distribution functions. The spectra of scattered light and $I_S/I_{AS}$ are similar to those calculated above. We cannot distinguish between exciton and the free-carrier cascades directly in our experiments. The excitonic description, however, seems straightforward due to enhanced (with respect to bulk materials) Coulomb effects in 1L-TMDs[1].

In conclusion, we investigated the light scattered and emitted by 1L-WSe$_2$ excited above the free-carrier gap. We detected a periodic modulation of phonon-assisted hot PL with a period ~15 meV both in S and AS. We measured the S and AS intensity evolution from 78 to 295 K. We explained these high-order processes using a cascade model where electrons (holes) make successive transitions between real states with a finite probability of radiative recombination at each step. The electron states in the $\Lambda$ valleys are intermediate states for efficient exciton relaxation. Our findings provide fundamental understanding of the initial steps of exciton relaxation in 1L-WSe$_2$, and can be used to design optoelectronic devices based on this material. Our approach can be extended also to other layered materials and their heterostructures, as well as to perovskites.

## Methods

**Raman and PL spectra fitting**. Supplementary Fig. 1a–c presents optical microscopy images of representative samples: (a) 1L-WSe$_2$ on Au and suspended 1L-WSe$_2$; (b) 1L-WSe$_2$ on SiO$_2$/Si; (c) 1L-WSe$_2$ on hBN. Representative PL spectra collected 295 K at 514 nm excitation are in Supplementary Fig. 1d, showing a peak ~1.65 eV related to the A-exciton resonance[1,12]. Supplementary Fig. 2a shows representative data fits. The spectrum, at 295 K for 532 nm excitation, is shown with black dots. Blue lorentzian functions are used to fit the Raman peaks (FWHM ~ 1–10 cm$^{-1}$). The residual spectral weight is also fitted with Lorentzians and results into the broader (FWHM ~ 50–80 cm$^{-1}$) peaks of the hot PL (red). A flat baseline is taken into account for the whole energy scale, since the background in the S spectral range increases due to the higher intensity of the S cascades compared to the AS ones. A fit is shown in Supplementary Fig. 2b. The Lorentzians overlap, creating an asymmetric broad background (indicated by yellow dashed lines in Supplementary Fig. 2b).

We now consider the thermally induced shift in the hot PL cascades of 1L-WSe$_2$ in the 78–295 K range. We analyze Pos(E′, A$_1'$), as shown in the normalized intensity spectra in Supplementary Fig. 2c. Pos(E′, A$_1'$) red shift as a function of T, Supplementary Fig. 2d. Although the overall T dependence is not linear[55], in the 78–295 K range we get:

$$k = (-0.00755 \pm 0.00083) \, \text{cm}^{-1}\text{K}^{-1}, \tag{11}$$

where $k$ is the slope. This corresponds to a shift of ~0.2 meV in the 78–295 K range. However, acoustic modes participate in the hot PL phonon cascades and their T dependence is weaker compared to optical phonons[66]. Thus ~0.2 meV is an upper limit of the expected shift of the hot PL cascades in this T range. The period ~15.42 ± 0.08 meV is quantified at 295 K by applying a linear fit in the position of the steps in the cascade in Fig. 2a, while the error bar corresponds to the standard error of the linear fit. This does not take into account other sources, such as the error in the dispersion of the grating, the fitting accuracy, etc, therefore the actual error bar is expected to be larger than 0.08 meV. Thus, although a shift of the order of less than one tenth of meV induced by acoustic phonons would be expected in this T range, it is very challenging to experimentally observe it in hot PL.

## Data availability
The data that support the findings of this study are available from the corresponding author upon request.

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

## Acknowledgements
We acknowledge funding from ANR 2d-vdW-Spin, ANR MagicValley, the Institut Universitaire de France, the RFBR and CNRS joint project 20-52-16303, EU Graphene Flagship, ERC Grants Hetero2D and GSYNCOR, EPSRC Grants EP/K01711X/1, EP/K017144/1, EP/N010345/1, EP/L016087/1. G.W. acknowledges the support from the National Science Foundation of China Grant No. 11904019 and Beijing Natural Science Foundation Grant No. Z190006.

## Author contributions
B.U., M.M.G., X.M., I.P. and A.C.F. conceived the project. I.P. and G.W. performed the experiments and acquired the data. I.P. analysed the results. I.P., E.M.A. and A.R.C. fabricated the samples. M.M.G. performed the theoretical studies. I.P., M.M.G., B.U., X.M. and A.C.F. wrote the manuscript. All authors discussed the results and commented on the manuscript at all stages.

## Competing interests
The authors declare no competing interests.
