## [Peer Review File · Nature Communications]

REVIEWER COMMENTS

Reviewer #1 (Remarks to the Author):

In this work, the authors have measured the interaction between phonon with the photo-excited charge carriers in 1L WSe₂. They found a series of peaks with a period of ~15meV (7 peaks Stokes and 5 anti-Stokes peaks) and they assign those peaks to phonon cascades. Based on the temperature dependency measurement, the assignment for Stokes and anti-Stokes peaks is convincing. The observed experimental phenomena is explained with an assisting of a theoretical model which is self-consistent. Although this phenomenon is very meaningful, the results seem to be too specific to this journal's readers. Besides, there are some scientific issues that affect the reliability of the discussion in the manuscript. In my opinion, the manuscript should be revised and maybe re-submitted to a physics journal.

1. As well known, curve fitting is quite subjective. Therefore, their accuracy and reliability are often questionable, especially when the peak intensities are also involved in the discussion. From the manuscript, we can see that some inaccuracy may occur in the curve fitting process. For example, in Fig. 1d, the baselines are crossing the experimental data points. In Fig. S1, the background of the experimental data are increasing at higher energy side, while a flat baseline is used for all energy range. This makes the accuracy of fitting results questionable.

2. In this manuscript, the emission of hot PL is assisted by the phonon, which seems reasonable. However, it is still needed to rule out other possibility for example, plasmon, excitons, etc.

3. The phonon energy should be dependent on temperature. Normally, the phonon will get soften at higher temperature. Thus, the periods of the Stokes (anti-Stokes) peaks should vary with the temperature. However, this is not the case as presented in the manuscript, can the author provide a reason for this?

4. There are several typos in the manuscript. Just to name one, in equation (5), both terms are given for the conditions of $j < 0$. It appears to me that the first term should be for $j > 0$. The authors should verify this.

Reviewer #2 (Remarks to the Author):

In the manuscript entitled "Efficient phonon cascades in a monolayer semiconductor", Ioannis Paradisanos et al. report the optical signature of phonon cascades via ultra-low cut-off frequency Raman spectroscopy. Their results are novel and convincing. The main findings are periodic peaks of hot PL, originating from strong carrier-phonon interactions during hot carrier thermalization processes. Explicit mechanism of hot carrier thermalization as well as exciton formation in a monolayer semiconductor has been in hot scientific debate, since ultrafast exciton built-up process is very challenging to be directly accessed in a time-resolved measurement, which in general requires few-cycle femtosecond laser pulses. The results presented in this manuscript timely provide new experimental insights regarding this topic, and more importantly in my opinion an elegant way to access the ultrafast exciton built-up process without performing an expensive time-resolved measurement. I believe this work fits well the readership of Nature Communications (NC) and would significantly contribute to a full understanding of optoelectronic properties of monolayer semiconductors. Hence, I think this work is in high quality and deserves to be published in NC. Before publication, the authors need to address several minor issues in the following:

(1) The authors made a comprehensive theoretical analysis and satisfactorily reproduce their experimental findings (periodic peaks of hot PL). However, for each order of hot PL peak, it can be noticed that there are 2, 3 or 4 corresponding Raman peaks (see Figure S1, red lines for hot PL and blue lines for Raman). How do these Raman peaks correlate with hot PL peaks? Why the number and amplitude of Raman peaks seems random? For now, it seems quite confusing here. Can the authors make some brief clarifications?

Fig3(a) present clearly the relations of phonon cascade and hot PL. Similarly, to further lift up the

impact of this manuscript, could the authors add a detailed cartoon for the Raman peaks as well ?

(2) The optical characterizations of the WSe₂ samples, such as optical microscope images, PL and absorptance, are absent in the current manuscript. These results sometimes are useful for others trying to repeat the Raman and hot PL results reported by the authors. Could the authors add these results in the supporting information?

(3) Why the authors choose WSe₂ ? Can the other TMDs, such as MoS₂ or WS₂, show the same optical signatures of phonon cascades. The authors might need to add some comments in the maintext, stating whether their findings are universal or limited to a specific material.

In summary, I would highly recommend the manuscript's publication in Nature Communications after the authors address my above concerns.

Reply to reviewer 1 ####
#####

We thank the reviewer for analyzing our work and for concisely summarizing the main findings. Below, we address the reviewer's comments.

Reviewer: *"..the assignment for Stokes and anti-Stokes peaks is convincing, the observed experimental phenomena is explained with an assisting of a theoretical model which is self-consistent. Although this phenomenon is very meaningful, the results seem to be too specific to this journal's readers. In my opinion, the manuscript should be revised and maybe re-submitted to a physics journal. "*

Reply: We are thankful for the positive feedback on our work. We would like to emphasize that given the approaching stage of industrialisation of layered materials, the topic of understanding carrier relaxation pathways **using elegant and accessible** approaches is timely for a broad range of applications, from photonics to optoelectronics or even quantum technology. We believe that our findings will be beneficial to the general interest readership of *Nature Communications* for future projects involving not only the colossal family of layered materials (1,036 easily exfoliable cases, N. Mounet et al., *Nature Nanotechnology*, 13, 2018) but also the rapidly growing field of layered heterostructures with an incredibly large number of combinations. Similar phenomena could be examined in the growing field of perovskite semiconductors with intricate carrier-phonon coupling (J Yang et al., *Nature Communications*, 8, 14120, 2017 & D. Zhao et al., *ACS Nano*, 13, 8, 8826, 2019) not reported so far. The observation and interpretation of the processes described in this work are highly desired not only for fundamental semiconductor physics but also for device applications. For instance, the inelastic scattering of excitons is one of the key factors leading to stimulated emission and lasing (H. Haug, *Journal of Applied Physics* 39, 4687, 1968 & K. Era and D. W. Langer, *Journal of Applied Physics* 42, 1021, 1971).

We have added now in the introduction the following sentence together with three relevant references: *"Our approach can in principle be extended also to a plethora of layered materials and their heterostructures as well as to other materials systems such as perovskites with intricate carrier-phonon coupling."* **Also, a similar sentence has been added in the conclusion.**

Q1. *"As well known, curve fitting is quite subjective. Therefore, their accuracy and reliability are often questionable, especially when the peak intensities are also involved in the discussion. From the manuscript, we can see that some inaccuracy may occur in the curve fitting process. For example, in Fig. 1d, the baselines are crossing the experimental data points. In Fig. S1, the background of the experimental data are increasing at higher energy side, while a flat baseline is used for all energy range. This makes the accuracy of fitting results questionable."*

Reply: We agree with the reviewer that several limitations can appear when attempting any curve fitting. Indeed, the accuracy and reliability of the fitting can be challenging especially when the intensity of the peaks is involved. Therefore, we need to explain in detail the process followed here.

First, we need to clarify that the fitting process in Fig. 1d was not applied to quantify any information on the absolute intensities but just to emphasize that the **position** of the peaks and **the relative intensity** between the steps in the cascades is independent of the underlying substrate. We confirm that the **absolute intensities** can vary between

measurements performed on different substrates. This is due to several effects – such as interference, backscattering collection efficiency, disorder, etc. – which can modify the total signal collected. This is exactly why we did not attempt any comparison in the intensity of the cascades between different substrates, we believe it is a complex problem with no significant contribution to the main idea of this work. However, the relative intensity modulation between the steps in the cascades as well as the position of the peaks is very reliable and reproducible. We should mention that only the 90nm SiO₂/Si cases (e.g. Fig.1b) have been considered for quantification and comparison with theory, after a careful fitting analysis.

We also need to comment on the background extraction of the experimental data, we thank the reviewer for pointing out this point. We attempted different approaches on how to treat the background and we concluded that a simple, flat baseline should be used for the whole energy scale since the background in the Stokes spectral range *naturally* increases and is not an artifact. The origin of this peculiar spectral response comes from the fact that the intensity of the cascades in the Stokes is always stronger compared to the anti-Stokes range, therefore the Lorentzian curves will naturally start to overlap, thus creating an asymmetric broad background between the Stokes and anti-Stokes spectral range.

So, the sum of Lorentzians gives the impression of a background signal, but in reality, the signal only consists of spectrally overlapping Lorentzians with different amplitude. This can be seen in Reply Figure R1: we show an example of separate Lorentzian functions (red dashed lines) with their sum result (black line) in the higher energy (anti-Stokes) and lower energy (Stokes). The energy separation of the peaks here was selected to be ~15meV. What appears to be “the background” (here roughly shown with yellow dashed lines) directly arises due to the overlap of the Lorentzians. Since the intensity of the steps in the cascade is always stronger in the Stokes range (phonon emission) compared to the anti-Stokes range (phonon absorption), the perceived background is naturally expected to be asymmetric and should not be eliminated. We should point out that the strongest background asymmetry is expected at T = 78 K where practically there is no contribution in the anti-Stokes range, shown in Fig. 2b. Finally, in our analysis and model we do not take into account the absolute experimental intensities but the *ratio* of the steps in the Stokes over anti-Stokes, shown in Fig. 3c. This further makes the analysis more reliable since it excludes several experimental artifacts such as slight variations in the laser power, etc.

Following the reviewer’s suggestions, the fitting details together with the following figure have now been added in part A of the supplementary information.

Q2. “In this manuscript, the emission of hot PL is assisted by the phonon, which seems reasonable. However, it is still needed to rule out other possibility for example, plasmon, excitons, etc.”

Reply: We thank the reviewer for this very interesting comment. Other possibilities such as plasmonic effects indeed should be excluded although the phonon-assisted hot PL cascades explain very well our temperature and wavelength dependent experimental observations. In fact, this is one of the reasons we performed experiments using various substrates besides the main 90nm SiO₂/Si substrates presented in our main analysis. In Fig. 1d the same effect is also demonstrated in WSe₂ on a dielectric material such as hBN, in a suspended sample but more importantly on a metallic, gold (Au) substrate. Possible plasmonic effects would modify the observed cascades in different dielectric environments because the decaying fields of surface plasmon waves propagating along interfaces are highly sensitive to the ambient refractive index variations. In addition, freely propagating shortwave (intervalley) plasmons should be directly detected in monolayers with a heavy electron-doping. In this case, a resonance is predicted in the THz absorption spectrum when the photon energy is about twice the spin splitting in the conduction band (D. V. Tuan et al., Phys. Rev. X 7, 041040, 2017), ruling out such a possibility here. Furthermore, the electron density of our WSe₂ samples on 90nm SiO₂/Si is at least one order of magnitude smaller than the expected doping regime ($\sim 10^{12}$ cm⁻¹, D. V. Tuan et al., Phys. Rev. X 7, 041040, 2017) for shortwave plasmons, therefore we believe the phonon-assisted hot PL cascades is the prominent driving mechanism.

In addition, contribution from specific ground and excited excitonic states can also be ruled out since 4 different excitation energies (all lying above the free-carrier gap, M. Goryca et al., Nature Communications 10, 1, 2019) demonstrate the same effect (Fig. 1c).

We have now added the following sentence together with the references in the “Experimental results and theory” section of the main text: “In addition, by

comparing results obtained from 1L-WSe₂ on different substrates and for different laser excitation energies, we argue that phonon assisted hot PL is the dominant mechanism for the observed cascades, whereas contributions from other excitations such as plasmons are negligible here.”

Q3. “The phonon energy should be dependent on temperature. Normally, the phonon will get soften at higher temperature. Thus, the periods of the Stokes (anti-Stoke) peaks should vary with the temperature. However, this is not the case as presented in the manuscript, can the author provide a reason for this?”

Reply: The reviewer is correct and we are grateful for this important remark, missing from the submitted version. Phonons are expected to soften (*i.e.* lowering energy/red shift) with increasing temperature due to the combined effects of thermal expansion and phonon anharmonicity. Therefore, the reviewer naturally expects that the extracted period of the step in the cascade process should be temperature dependent. Let us now quantify the thermally induced shift in the hot PL cascades of WSe₂ monolayers in the temperature range between 78K and 295K. We have analyzed the position of the Raman active and degenerate first order E' & A'₁ optical phonons, as shown in the normalized intensity spectra below in Fig. R2 (left panel). The position indeed red shifts as a function of temperature, therefore we extract the absolute values and plot them as a function of temperature, shown in the right figure. Although the temperature dependence is not linear (Z. Li et al, Nano Res., 13(2) 591, 2020), the range between 78K and 295K is small enough to be approximated linearly. By applying a linear fit, we extract a slope of $(-0.00755 \pm 0.00083) \text{cm}^{-1}/\text{K}$. Converting this value from cm^{-1} to meV and considering that our temperature changes from 78K to 295K, we would expect a change of **~0.2 meV** (Fig. R2, right panel). However, in this consideration we took into account thermal shifts of optical phonons **and not acoustic ones**. Only the latter are related to the hot PL phonon cascades and they do not change notably with temperature compared to the optical frequencies (Fig.1 of A. Mobaraki et al., Phys. Rev. B, 100, 035402, 2019). As a result, the value of **~0.2 meV** we found before determines an overestimated upper limit of the expected shift in this range of temperatures. Additionally, the period of $(15.42 \pm 0.08) \text{meV}$ we present in the main manuscript has been experimentally quantified at 295K by applying a linear fit in the position of the steps in the cascade while the error bar corresponds to the standard error of this fit. This error does not take into account other error sources, such as possible grating dispersion deviation, Lorentzian fitting accuracy, etc. Therefore, we estimate that the error value can be slightly larger than 0.08 meV. We can thus conclude that although a shift on the order of less than one tenth of meV induced by acoustic phonons would be expected in this range of temperatures, it is very challenging to experimentally observe it in the hot PL cascades.

We included this very helpful remark **stimulated by the reviewer's question** in the main text: “Thermal effects are expected to modify also the phonon energies. However, within this range of temperatures we did not observe any measurable shifts in the position of the hot PL peaks because the shifts induced by acoustic phonons are smaller compared to our experimental error (see Supplementary Part A).” We also added the figure R2 in the supplementary information, as Figure S2 together with the details of the analysis.

Reply Fig. R2. Left: normalized Raman intensity of the first order degenerate E' , A'_1 at different temperatures. Right: position of E' , A'_1 as a function of temperature and the applied linear fit.

Q4. "There are several typos in the manuscript. Just to name one, in equation (5), both terms are given for the conditions of $j < 0$. It appears to me that the first term should be for $j > 0$. The authors should verify this."

Reply: We appreciate the detailed examination of the text from the reviewer. We have now corrected the typos.

Reply to reviewer 2 ####
#####

Reviewer 2 *“Their results are novel and convincing and timely provide new experimental insights regarding this topic, and more importantly in my opinion an elegant way to access the ultrafast exciton built-up process without performing an expensive time-resolved measurement. I believe this work fits well the readership of Nature Communications (NC) and would significantly contribute to a full understanding of optoelectronics properties of monolayer semiconductors. Hence, I think this work is in high quality and deserves to be published in NC.”*

Reply: We greatly appreciate the very positive evaluation of our manuscript and the concise summary of our main findings. Below we provide our detailed reply to the questions:

Q1: *“The authors made a comprehensive theoretical analysis and satisfactorily reproduce their experimntal findings (periodic peaks of hot PL). However, for each order of hot PL peak, it can be noticed that there are 2, 3 or 4 corresponding Raman peaks (see Figure S1, red lines for hot PL and blue lines for Raman). How does these Raman peaks correlate with hot PL peaks? Why the number and amplitude of Raman peaks seems random? For now, it seems quite confusing here. Can the authors make some brief clarifications? Fig3(a) present clearly the relations of phonon cascade and hot PL. Similarly, to further lift up the impact of this manuscript, could the authors add a detailed cartoon for the Raman peaks as well ?”*

Reply: We thank the reviewer for raising this very interesting point. In all cases studied we applied a complete fit, *i.e.* including both the hot PL emission and the spectrally sharper Raman peaks. This complete fit gives a reliable determination of the quantification of the intensity values of the hot PL peaks.

In brief: The hot PL is a result of generating an electronic excitation at this particular energy. So, the Raman signal coinciding with the hot PL energy is therefore resonant Raman scattering, *i.e.* the outgoing Raman light for the sharp peaks is in resonance with a real electronic state.

In detail: Regarding the appearing sharp Raman peaks, we do not believe their emergence is random; Let us first take a closer look in their correlation with the hot PL cascades. In the supplementary (new) Fig. S2 we present an example of a fitted spectrum at 295K. Indeed, it seems that sharp Raman peaks are grouped solely within the spectral range of the cascade steps. We can see that for the first step ($j = \pm 1$) there are three peaks appearing symmetrically both in the Stokes and anti-Stokes. Their frequencies lie in 96.5cm^{-1} , 118.5cm^{-1} and 138.2cm^{-1} . Following the calculations by Z. Jei et al. (Phys. Rev. B, 90, 045422, 2014), these peaks could be attributed to the TA(Λ), LA(Λ) and LA(K) phonon branches, respectively.

If we now move to the second step ($j = \pm 2$), we see that there are five Raman peaks. The second step in the cascade coincides with the two strong, degenerate, first order E' & A' optical phonons at 250cm^{-1} (also shown in Fig.1a of the main manuscript) at the centre of the Brillouin zone (Γ). Above and below 250cm^{-1} the peaks associated with

the second-order processes involving two phonons within the first Brillouin zone or involving a phonon and a defect emerge. The two peaks located at 258cm^{-1} and 263cm^{-1} correspond to the $2\text{LA}(\text{M})$ phonon branch at the M point (overtone of the $\text{LA}(\text{M})$) and to an A-symmetry optic branch at the M point (E. del Corro et al., ACS Nano, 8, 9, 9629, 2014). Phonon modes of the E-symmetry optical branch at the K and M points of the Brillouin zone, could explain the rest two peaks at 219cm^{-1} and 243cm^{-1} , respectively.

In the third step of the cascade ($j = \pm 3$), three sharp peaks appear at 361cm^{-1} , 375cm^{-1} and 398cm^{-1} . The origin of these second order features is not yet entirely clear. K. Goasa et al. (Appl. Phys. Lett., 104, 092106, 2014) propose the participation of transversal acoustic modes in the existence of these resonant bands while W. Zhao et al. (Nanoscale, 5, 9677, 2013) assign them to a combination of acoustic and optical phonons at different high symmetry points. H. Li et al. (Small, 9, 11, 1974, 2013), assign these modes to 2E_{1g} , $\text{A}_{1g}+\text{LA}$ and $2\text{A}_{1g}-\text{LA}$, respectively.

Finally, in the fourth step of the cascade ($j = \pm 4$), two sharp peaks appear. The one at 520cm^{-1} originates from the strong vibrational mode of the underlying silicon substrate (P. A. Temple et al., Phys. Rev. B 7, 3685, 1973) whereas the assignment of the mode at 495cm^{-1} is not yet clear.

For the next steps in the cascade, no Raman peaks are observed. In general, during this resonant process, not only one-phonon modes but also higher order multiphonon modes exhibit enhancement whose strengths decrease as a function of the step in the cascade (Y. Peter and M. Cardona, Springer, 2010). Therefore, together with the hot PL, it is possible that first and second order Raman processes coinciding in energy with the hot PL are also enhanced. The amplitude of the different Raman modes observed is related to several factors, such as whether it is a first or higher order process, the resonant energy, the Raman cross section, etc. As a result, their intensity interpretation is a very complex problem which may be very interesting for future studies. Although we understand the point of the reviewer to include a cartoon of the Raman peaks too, we are afraid that the complexity of the large number of different Raman modes – together with their higher order combinations, some of them not fully verified yet – might confuse and disorient the reader from the key message of this work.

We have included a statement together with new references regarding the sharp Raman peaks in the main text: “..while weaker Raman peaks are also observed between 90cm^{-1} (11meV) and 500cm^{-1} (62meV) (see Supplementary Part A) and have been identified elsewhere.”

Q2: “The optical characterizations of the WSe_2 samples, such as optical microscope images, PL and absorptance, are absent in the current manuscript. These results sometimes are useful for others trying to repeat the Raman and hot PL results reported by the authors. Could the authors add these results in the supporting information?”

Reply: We thank the reviewer suggesting these improvements to the manuscript. We have now included the optical microscope images, as well as the corresponding PL spectra of the different samples in the new Supplementary Fig. S1 and **we added the following sentence in the main text:** “..optical microscope images and PL characterisation shown in Supplementary Part A.”

Q3: *“Why the authors choose WSe₂ ? Can the other TMDs, such as MoS₂ or WS₂, show the same optical signatures of phonon cascades. The authors might need to add some comments in the maintext, stating whether their findings are universal or limited to a specific material.”*

Reply: We appreciate this very interesting argument. We attempted to collect optical signatures also from other TMDs, such as 1L-MoS₂ and 1L-WS₂. We did not observe any noticeable periodic emission, similar to reported here for WSe₂ monolayers. We believe this observation is in line with our proposed scenario, as the phonon scattering mechanism is linked to the particular bandstructure of the material and the involvement of Λ -valleys as intermediate states for efficient relaxation in monolayer WSe₂. The K- Λ energy difference in WSe₂ is estimated to be 35meV, as compared to 81meV, 137meV and 207meV for WS₂, MoSe₂ and MoS₂, respectively (Ref. 55 in the main text, A. Kormányos et al., 2D Mater., 2, 049501, 2015).

The following sentence has now been added in the main text: *“Note that we did not observe any noticeable periodic emission for 1L-MoS₂ and 1L-WS₂ samples investigated. This observation is in support of our proposed scenario, as the phonon scattering mechanism is linked to the particular bandstructure of 1L-WSe₂.”*

Reviewer #1 (Remarks to the Author):

The manuscript has been well revised. My previous concerns as well as the comments have been treated nicely by the authors in their revised manuscript and their response letter. The quality and the readership have been highly improved for the revised version of the manuscript. I now recommend its publication in Nature Communications in its current form.

Reviewer #2 (Remarks to the Author):

In the revised manuscript and supporting information, the authors have satisfactorily addressed all my concerns. At this point, I highly recommend its publication in Nature Communications. Congratulations!

Referee 1 states:

“The manuscript has been well revised. My previous concerns as well as the comments have been treated nicely by the authors in their revised manuscript and their response letter. The quality and the readership have been highly improved for the revised version of the manuscript. I now recommend its publication in Nature Communications in its current form.”

Referee 2 states:

“In the revised manuscript and supporting information, the authors have satisfactorily addressed all my concerns. At this point, I highly recommend its publication in Nature Communications. Congratulations!”

We thank the referees for the positive comments and for the detailed evaluation that resulted in the improved version of our manuscript.